# Detection of Sub-Nanomolar Concentration of Trypsin by Thickness-Shear Mode Acoustic Biosensor and Spectrophotometry

**DOI:** 10.3390/bios11040117

**Published:** 2021-04-11

**Authors:** Ivan Piovarci, Sopio Melikishvili, Marek Tatarko, Tibor Hianik, Michael Thompson

**Affiliations:** 1Department of Nuclear Physics and Biophysics, Faculty of Mathematics, Physics and Informatics, Comenius University, Mlynska dolina F1, 84248 Bratislava, Slovakia; piovarci6@uniba.sk (I.P.); s.melikishvili@gmail.com (S.M.); marek.tatarko@fmph.uniba.sk (M.T.); 2Lash Miller Laboratories, Department of Chemistry, University of Toronto, Toronto, ON M5S 3H6, Canada

**Keywords:** trypsin, β-casein, AuNPs, acoustic wave biosensor, colorimetric assay

## Abstract

The determination of protease activity is very important for disease diagnosis, drug development, and quality and safety assurance for dairy products. Therefore, the development of low-cost and sensitive methods for assessing protease activity is crucial. We report two approaches for monitoring protease activity: in a volume and at surface, via colorimetric and acoustic wave-based biosensors operated in the thickness-shear mode (TSM), respectively. The TSM sensor was based on a β-casein substrate immobilized on a piezoelectric quartz crystal transducer. After an enzymatic reaction with trypsin, it cleaved the surface-bound β-casein, which increased the resonant frequency of the crystal. The limit of detection (LOD) was 0.48 ± 0.08 nM. A label-free colorimetric assay for trypsin detection has also been performed using β-casein and 6-mercaptohexanol (MCH) functionalized gold nanoparticles (AuNPs/MCH-β-casein). Due to the trypsin cleavage of β-casein, the gold nanoparticles lost shelter, and MCH increased the attractive force between the modified AuNPs. Consequently, AuNPs aggregated, and the red shift of the absorption spectra was observed. Spectrophotometric assay enabled an LOD of 0.42 ± 0.03 nM. The Michaelis–Menten constant, K_M_, for reverse enzyme reaction has also been estimated by both methods. This value for the colorimetric assay (0.56 ± 0.10 nM) is lower in comparison with those for the TSM sensor (0.92 ± 0.44 nM). This is likely due to the better access of the trypsin to the β-casein substrate at the surface of AuNPs in comparison with those at the TSM transducer.

## 1. Introduction

Peptidases, more frequently referred to as proteases, are a group of enzymes that irreversibly hydrolyze a peptide bond in an amino acid sequence through the nucleophilic attack and subsequent hydrolysis of a tetrahedral intermediate. They play critical roles in biological and physiological processes such as blood clotting, digestion, and a variety of cellular activities [1,2]. Proteases are highly involved in the dairy industry as well, where their activity is directly linked to the shelf life of dairy products [3]. Owing to their specificity, protease activity-based biosensors are used in various diseases diagnostics [4,5,6]. For example, pancreatic diseases such as cystic fibrosis, acute pancreatitis, or the acute phase of chronic pancreatitis are associated with the increased trypsin level of 2.1–71.42 nM in the serum of patients [7,8]. In the healthy physiological condition, the concentration of trypsin varies in magnitude. Additionally, levels of trypsin differ between serum and intestinal levels. For serum levels for fasting individuals, the concentration of trypsin was measured from 4 to 20 nM [9,10]. The intestinal level of trypsin depends on the location in the intestine and ranges from 4 to 30 μM, which is much higher than in serum [11].

Moreover, the inhibitors of these proteases are successfully employed as therapeutic agents [2,12,13].

Trypsin is an extremely important serine protease of the chymotrypsin family. It is produced in the pancreas and it plays crucial roles in the small intestine. Trypsin catalyzes the hydrolysis of consumed proteins and activates protease proenzymes as part of the digestive system. It is highly specific toward the cleavage of peptide bonds at the carboxyl side of lysine or arginine. Trypsin is often used as a model protease because it is inexpensive and readily available [14,15,16]. Standard assays for the detection of proteases such as trypsin usually utilize fluorogenic and chromogenic substrates. Those assays are useful, practical, and highly sensitive. However, spectroscopic assays are incapable of measuring protease activity in highly colored and turbid samples such as cells, tissue lysates, or milk. Therefore, the development of a new label-free method for detecting protease activity without interruption from impurity inclusions is needed [1,15,17].

The thickness-shear mode (TSM) acoustic wave biosensor may present an attractive platform for the development of cost-effective and highly sensitive techniques for trypsin detection. The use of TSM devices is a well-known method for the detection of mass changes due to depositions or chemical/biochemical reactions on its surface. It is also an established method for detecting changes in the viscoelastic properties of the contacting material. Therefore, the TSM biosensor is a sensitive tool for the study of molecular interactions on surfaces [18]. Moreover, the coupling of a flow injection analysis (FIA) system to a TSM sensor device permits the monitoring of kinetic processes that take place at the surface of the sensor [19]. The TSM device applies a high-frequency AC voltage across an AT-cut quartz crystal on which, due to the piezoelectric effect, an acoustic shear wave is generated and propagated through the sensing layer perpendicular to the surface of the crystal [20]. It has a low noise level and higher Q-factor in clinical liquids such as tissue fluids and serum. Compared to other common biosensing technologies, TSM electroacoustic resonators have the combined advantages of high sensitivity and low cost, label-free detection of analyte, and simple operation without the requirement of bulky detection systems [21]. Moreover, in contrast with traditional quartz crystal microbalance (QCM) techniques, the analysis of complex impedance spectra allows for the receipt of information about changes in the properties of layers even with the adsorption of relatively small molecules that do not contribute to the mass but only to the viscoelastic properties of the layer [22]. The multi-harmonic QCM method has previously been applied for the detection of plasmin and trypsin at the surface of β-casein layers [23]. This method allows the detection of these proteases at the sub-nM level. However, the possible contribution of viscoelastic effects has not been analyzed.

In addition to the acoustic methods also the colorimetric assay based on gold nanoparticles (AuNPs) for protease detection is of increased interest. AuNPs have attracted tremendous interest because of their optical and electronic properties, which are tunable by changing the size, shape, surface chemistry, or aggregation state. Colloidal AuNPs have a distinctive red color, which arises from the tiny dimensions of the AuNPs. The changes in the UV–vis spectra of the resultant colloids are measured to investigate the size effect of AuNPs on the surface plasmon resonance (SPR). Interestingly, the red color of citrate-stabilized AuNPs turns to blue when they are aggregated [24]. This approach has been widely applied to various methods for colorimetric detection of analytes via the aggregation of AuNPs [25], including those of trypsin detection [26].

In this work, we designed an analytical method based on the TSM biosensor for the real-time and label-free detection of trypsin. Using TSM frequency responses, we studied the assembly and stability of self-assembled β-casein layers on a quartz crystal electrode and measured the dynamics of TSM response and changes in motional resistance during casein cleaving by the protease.

Additionally, we compared the sensitivity of the TSM method with another label-free assay of the protease activity by employing AuNPs coated by β-casein and 6-mercaptohexanol (MCH). Unlike the surface-sensitive TSM biosensor, the suggested approach was volume-sensitive, thus allowing us to monitor tryptic activity in the reaction mixture. We used an approach developed by Chuang et al. [26]. However, instead of gelatin, β-casein was used as a substrate for trypsin. β-casein adsorbed on AuNPs kept the modified nanoparticles stably suspended in solution.

Considering the results obtained, we believe that the proposed approaches constitute rapid, cost-efficient, sensitive and useful tools for protease analysis. This paper is an extension of a conference paper published in the 1st International Electronic Conference on Biosensors [27].

## 2. Materials and Methods

### 2.1. Reagents

Ultrapure water obtained by reverse osmosis (Thermo Scientific, Waltham, MA, USA, ρ = 18.2 MΩ cm) was used for the preparation of all aqueous solutions. As a medium, 10 mM, pH 7.4 phosphate-buffered saline (PBS) was used (10 mM Na_2_HPO_4_, 2 mM KH_2_PO_4_, 2.7 mM KCl and 137 mM NaCl), prepared from tablets (Sigma-Aldrich, Darmstadt, Germany, Cat. No. P4417). In the experiments, trypsin from bovine pancreas (≥90%, ≥7500 BAEE units/mg solid, Sigma-Aldrich, Darmstadt, Germany, Cat. No. T9201) served as a model protease. The concentration of stock bovine β-casein (≥98%, Sigma-Aldrich, Darmstadt, Germany, Cat. No. C6905) solutions, prepared in PBS, was 0.5 mg/mL. 11-Mercaptoundecanoic acid (MUA, Sigma-Aldrich, Cat. No. 450561), *N*-(3-dimethylaminopropyl)-*N*′-ethylcarbodiimide (EDC, ≥98%, Sigma-Aldrich, Cat. No. E6383), and N–Hydroxysuccinimide (NHS, Sigma-Aldrich, Darmstadt, Germany, Cat. No. 130672) were employed for β-casein immobilization. The chemicals needed to prepare the gold nanoparticles, such as auric acid (HAuCl_4_), sodium citrate, and 6-mercapto-1-hexanol (MCH), were purchased from Sigma-Aldrich (Darmstadt, Germany). All experiments were carried out at 20 °C.

### 2.2. Cleaning and Modification of Gold Electrode-Coated Quartz Crystals

Symmetric gold electrode-coated quartz discs (Total Frequency Control, Storrington, UK, working area, 0.2 cm^2^) with a fundamental frequency of 8 MHz were cleaned in a basic Piranha solution (29% NH_3_, 30% H_2_O and H_2_O_2_ with volumetric 1:5:1 ratio, respectively) for 25 min. After this treatment, the crystals were washed three times with deionized water and stored in ethanol. After drying in a flow of nitrogen, the TSM crystals were immersed in 2 mM MUA and were incubated for 16 h to form a self-assembled monolayer. After this step, the crystals were rinsed several times with deionized water and dried under nitrogen, followed by incubation for 20 min in a 20 mM EDC and 50 mM NHS mixture in order to activate the carboxylic groups of MUA for the further immobilization of bovine β-casein on the gold electrode of the quartz sensor. The scheme of modification of the TSM transducer as well as the cleavage of β-casein by trypsin is shown in Figure 1.

### 2.3. TSM Measurements

AT-cut 8.0 MHz gold electrode-coated quartz crystals, modified on one side by MUA with activated carboxylic groups by NHS/EDC as described above, were incorporated into a home-built flow-through thickness shearing mode (TSM) acoustic wave device sensor system. The setup and general configuration of the flow-through system is described in reference [19]. One side of the crystal was exposed to liquid, the other one was exposed to air. The liquid was introduced using a syringe pump (Genie Plus, Torrington, CT, USA). Runs were performed with the crystals in the vertical position and at ambient temperature (approximately 20 °C). The modified crystal was secured in the holder using two O-rings. The gold electrodes were kept in contact with the gold leads in the holder. Resonance frequency, f, and motional resistance, R_m_, were determined based on the Butterworth–van Dyke (BVD) model of a quartz crystal resonator [19]. The resonant frequency represents the energy storage and reflects the mass changes of the oscillating layer, while R_m_ is related to the dissipation of energy and provides evidence of changes in the shearing viscosity of the layer [22]. The measuring procedure was as follows. Each slide was flushed through with PBS at a flow rate of 50 μL/min until a stable baseline was achieved (45 min), using the flow-through injection system. This step was necessary to remove any weakly adsorbed molecules at the surface of the TSM transducer. Next, the pump was momentarily stopped. The β-casein solution (0.5 mg/mL in PBS) was slowly introduced to the sample, while the PBS was exchanged out in order to minimize pressure effects to the system. β-casein was introduced at a rate of 50 μL/min for approximately 45 min. Once again, the pump was momentarily stopped, and the sample input tube was slowly placed back into the PBS solution. The PBS was re-introduced at a rate of 50 μL/min to remove any loosely bound casein until a stable baseline was achieved. Changes of the resonant frequency and motional resistance were recorded. For proteolysis measurements, solutions with various concentrations of trypsin in PBS (0.1, 0.5, 1, 5, 10, and 20 nM) were flowed over TSM crystals with an immobilized β-casein layer at a flow rate of 50 μL/min. Trypsin and β-casein solutions were freshly prepared before each experiment.

### 2.4. Synthesis and Modification of AuNPs

AuNPs were prepared using a modified citrate method described in reference [28]. Briefly, 100 mL of HAuCl_4_ (0.01%) was heated to boiling under vigorous stirring, which was followed by the addition of 5 mL of sodium tris-citrate solution (1%). The solution was left boiling while stirring until it turned a deep red. Then, we let the AuNPs solution cool down and stored it in the dark. In order to modify the gold nanoparticles with casein, we added 2 mL of 0.1 mg/mL β-casein to 18 mL of the AuNPs solution. After 2 h of incubation at room temperature without stirring, the gold nanoparticles were further incubated with 200 µL of 1 mM MCH overnight for approximately 18 h. The scheme of modification of AuNPs is showed in Figure 2.

### 2.5. Sprectrophotometric Assay

For the colorimetric assay, we prepared 0.95 mL of AuNPs. Trypsin was dissolved in deionized water, and 0.05 mL of trypsin from the stock solution was added to each cuvette. The concentration of trypsin in cuvettes was 0.1, 0.5, 1, 5, and 10 nM at 1 mL total volume of solution. We also used a reference cuvette where only 0.05 mL of protease-free water was added to the AuNPs solution. We measured the spectra of the AuNPs before trypsin addition (0 min), just after trypsin addition (approximately 1 s), and then every 15 min up to 60 min. The measurement was repeated 3 times. We multiplied the value of absorbance at time t = 0 by the dilution factor to correct the changes in absorbance intensity caused by the initial protease addition. Absorbance was measured by UV-1700 spectrophotometer at a temperature of around 20 °C and in the wavelength range of 220–800 nm (Shimadzu, Kyoto, Japan).

### 2.6. Analysis of Casein Adsorption and Hydrolysis Processes

The surface concentration (*Γ_QCM_*, ng/cm^2^) of the adsorbed β-casein layer on the TSM transducer was determined by a modified Sauerbrey Equation (1) as follows:(1)$$\Gamma_{QCM}=\frac{-A\sqrt{µ\rho }\Delta f}{2f_{0}^{2}},$$
where A is the area of the electrode, *ρ* = 2.648 g/cm^3^ is the density of quartz, μ = 2.947 × 10^11^ g/cms^2^ is the shear modulus of AT-cut crystal, and *f*_0_ is fundamental resonant frequency [29]. The Sauerbrey equation is strongly valid for thin rigid films at the surface of quartz crystal in vacuum. However, in a liquid, the viscoelastic contribution can affect the frequency changes. Through analysis of the motional resistance, R_m_, it is possible to estimate whether the mass or viscosity is dominant in frequency changes. It has been shown that the slope of |Δf/ΔR_m_| can be used for quantitative estimation whether the changes in frequency can be attributed to mass or to viscosity effects. For ideal rigid films, the ΔR_m_ values are practically zero. This means that |Δf/ΔR_m_| parameters higher than a certain critical value can be assigned to the mass effect [30]. According to the calculations made in ref. [30] for the AT cut quartz crystal with fundamental frequency *f*_0_ = 8 MHz, |Δf/ΔR_m_| = 10.37 Hz/Ω.

The frequency changes following the addition of the trypsin were normalized to the changes of the resonant frequency caused by adsorption of the β-casein at the surface of TSM crystal. This allowed consideration of a possible variation in the properties of the β-casein layers that were subsequently cleaved by trypsin. The normalized frequency changes were expressed as Δf_N_ = (Δf_TRY_/Δf_casein_) × 100(%), where Δf_TRY_ are changes in frequency following the addition of trypsin at certain concentration of the protease and Δf_casein_ are changes in frequency caused by the formation of a β-casein layer.

An inverse Michaelis–Menten (MM) model [31] was used to describe the dependence of the normalized frequency changes vs. concentration of trypsin at fixed concentration of the β-casein at the surface of TSM transducer:(2)$$\Delta f_{N}={\left(\Delta f_{N}\right)}_{max}\frac{C_{TRY}}{K_{M}+C_{TRY}}$$
where (Δ*f_N_*)*_max_* is the maximal change of the frequency that corresponds to the maximum rate of enzyme reaction achieved by the system happening at saturating enzyme concentration, *C_TRY_* is the concentration of trypsin, *K_M_* is the reverse Michalis–Menten constant that is equal to the trypsin concentration that achieves half of maximum rate. The hydrolysis of β-casein in a volume was modeled with an inverse MM Equation (3) as well
(3)$$\frac{A_{0}-A_{15}}{A_{0}}100=v_{max}\frac{C_{TRY}}{K_{M}+C_{TRY}},$$
where *A*_0_ is the absorbance of AuNPs before exposure to trypsin, *A*_15_ is the absorbance after 15 min of exposure to trypsin, and $v_{max}$ = [100 × (*A*_0_ − *A*_15_)/*A*_0_]*_max_* represents the maximum rate achieved by the system.

### 2.7. Data Analysis

Origin version 7.5 software (Microcal Software Inc., Northampton, MA, USA) was used for curve-fitting and data analysis. Data were obtained from a minimum of 3 independent experiments.

## 3. Results and Discussion

### 3.1. Development of Acoustic Biosensor for the Detection of Trypsin Activity at Surfaces

For the detection of trypsin activity at surfaces, it is crucial to optimize the methods of preparation of the protein layers that serve as a substrate for the protease of interest. The preparation of the protein layers on the surface of the transducers is a common application of acoustic biosensors. For instance, the preparation of casein layers is attractive for future applications in the pharmaceutical and food industries [32].

In this study, we have monitored the activity of trypsin at various concentrations (from 0.1 to 20 nM) in the hydrolysis of a β-casein layer immobilized onto a gold surface by a carboxylate terminated self-assembled monolayer (SAM) of MUA using a TSM technique. MUA strongly binds to gold through thiol groups in a high level of molecular dimension order, forming a stable SAM [33]. The formation of the SAM itself enables the coupling of activated carboxylic groups with free amino groups in the β-casein, which is an effective method for immobilizing proteins on a gold surface [34,35,36].

Figure 3 illustrates typical kinetic changes of the frequency, Δf, and motional resistance, ΔR_m_, obtained during the TSM experiment. The TSM crystal covered by the MUA layer activated by EDC/NHS established in a flow cell has been first washed by PBS. As soon as the stable baseline was established, the β-casein dissolved in PBS in a concentration of 0.5 mg/mL has been added. The sharp drop of the resonant frequency and an increase of the motional resistance were observed, indicating the adsorption of the β-casein to the quartz crystal/liquid interface. The washing of the surface by PBS resulted in only a slight increase of the frequency, which is evidence of removal of weakly adsorbed β-casein molecules from the surface. Thus, the frequency did not recover to the original value obtained when the crystals were exposed to the buffer. This suggests that there were two modes of casein binding to the MUA surface, a tightly bound layer and a weakly bound layer, and that only loosely bound casein layers were removed during the PBS washing [37]. Since the increase in resonant frequency after PBS washing was so small, we can speculate that β-casein adsorbed on the MUA formed a stable immobilized layer, which makes this result attractive for its potential applications in biosensors for the detection of protease activity.

The resulting frequency shift after the adsorption of the β-casein to the surfaces of the crystal was around −199.43 Hz. Furthermore, the buffer was changed to a 20 nM trypsin solution. The frequency increased asymptotically to reach a stable value, indicating that the proteolysis process occurred. Washing of the surface by PBS did not result in significant changes of frequency and motional resistance, which is evidence that the cleaved peptide residues were removed from the surface in a flow mode during the application of trypsin. The kinetics of the changes of the resonant frequency and motional resistance were recorded for different trypsin concentrations, each one with a new quartz crystal and a newly adsorbed β-casein layer.

Earlier works indicated that the Sauerbrey Equation (1) can be applied to obtain a rough estimate for the surface concentration of the adsorbed β-casein layer [23,38,39], which is valid only for the specific case of a crystal being loaded with rigid, well-adhered layers in air with a minor contribution to the surface viscosity [19,40,41]. As we mentioned in Section 2.6, the contribution of viscosity into the frequency changes can be estimated from the ratio |Δf/ΔR_m_|. At the highest concentration of trypsin (20 nM) studied and at the steady-state conditions (Figure 1), |Δf/ΔR_m_| = 199.43 Hz/7.4 Ω = 26.95 Hz/Ω. This value is much higher than the threshold value (10.37 Hz/Ω). This means that the changes of frequency are related mainly to the changes of the mass.

Therefore, with an awareness of the limitations stated above, Equation (1) can be used to estimate the amount of proteins on the surface (Γ_QCM_, ng/cm^2^) [38]. The average value of the frequency shift after the adsorption of the β-casein to hydrophilic surfaces was −165.26 ± 47.7 Hz. Using this value, as well as A = 0.2 cm^2^ for the area of the electrode of an AT-cut quartz crystal (f_0_ = 8 MHz fundamental resonant frequency), a surface concentration of 228.1 ± 65.8 ng/cm^2^ was obtained for β-casein.

This is in good agreement with earlier experimental works based on ellipsometry that reported 200–300 ng/cm^2^ for a full-coverage monolayer of β-casein [42,43,44]. Furthermore, QCM studies by Tatarko et al. estimated a mass density of 350 ng/cm^2^ for the immobilized β-casein monolayer [23]. These results support the interpretation that the surface concentration of β-casein obtained by TSM measurements corresponds to monolayer formation.

Based on the kinetic curves obtained for the concentration range of trypsin 0.1–20 nM, we prepared a plot of the frequency and motional resistance changes as a function of trypsin concentration (Figure 4). It can be seen that the frequency changes increase with increasing the trypsin concentration and started to saturate at C_TRY_ > 10 nM. In contrast with frequency, R_m_ decreased with increasing the concentration of the protease, which is evidence of dominant mass changes.

For practical purposes, for the detection of trypsin in food or in other biological samples such as blood or blood plasma, it is convenient to analyze the effect of trypsin on the cleavage of β-casein by changes of resonant frequency of the quartz crystal under steady-state conditions. In order to minimize the effect of variation of the properties of β-casein layers at the monolayer of 11-mercaptoundecanoic acid (MUA) on the resonant frequency, we plotted the normalized frequency changes: Δf_N_ = (Δf_TRY_/Δf_casein_) × 100% vs. concentration of trypsin, C_TRY_ (Δf_TRY_ is the frequency change corresponded to the cleavage of β-casein layer after incubation with a certain concentration of trypsin and Δf_casein_ is the frequency changes corresponded to the adsorption of β-casein at the MUA layer before trypsin addition). This dependence shown on Figure 5 can be fitted by the Langmuir isotherm (see Section 2.6 and Equation (2)).

The fitting of calibration plots yielded (Δf_N_)_max_ = 70.36 ± 4.60 and K_M_ =0.92 ± 0.44 nM. The limit of detection (LOD) has been determined from the linear part of the dependence presented in Figure 5 using the 3.3(SD)/S rule (SD is standard deviation at the lowest concentration of trypsin, S is the slop of the linear dependence) as LOD = 0.48 ± 0.08 nM. Thus, in the presence of 20 nM trypsin, almost 70% of the casein layer is removed due to protease cleavage. This value is close to the maximum cleavage obtained by fitting the Langmuir isotherm. It can be assumed that due to the restricted access of the trypsin to the casein layer at the surface of the TSM transducer, around 30% of the casein remained at the surface after protease cleavage.

β-casein interacts with the immobilized MUA layer preferably with N-terminus. This part of the protein contains most of the charge [45]. It also contains numerous free amino groups that amino-reactive MUA can bind. β-casein is composed of 209 amino acids starting with arginine at the N-end (Arg1) [46]. The immediate binding of Arg1 to MUA is possible. Following the addition of trypsin, the cleavage of available peptide bonds toward the C-terminus of lysine and partially arginine occurs. These cleavage sites for trypsin are mostly identical to that of the plasmin [47]. The only unique cleavage site for trypsin is located between Arg202-Gly203, near the C-terminus [48]. The most common hydrolysis takes place at Lys28-Lys29, Lys105-His106, and Lys107-Glu108 with the subsequent cleavage at Lys97-Ala98, Lys99-Glu100, and Lys113-Tyr114 [49]. The cleavage of these peptide bonds should cause release of the β-casein fragments (or so called γ-casein fragments) that corresponds to up to 88% of the β-casein molecular weight. This ratio can be affected by the β-casein assembly on the MUA layer and thus the availability of such bonds to the trypsin. Considering that approximately 70% of casein fragments are released from the sensing, we can speculate that the closest site for its cleavage by trypsin at the MUA layer is probably after Lys48. According to the ExPASy Peptide Cutter tool [50], the cleavage of β-casein by trypsin at Lys48 is highly probable.

In the paper by Chen et al. [51], the detection of trypsin activity based on the electrochemical method has been reported. They applied a gold working electrode modified with a short peptide substrate conjugated with graphene oxide (GO) and the thionine redox label. The incubation of the sensor with trypsin for 2 h resulted in cleavage of the peptide substrate, removal of the redox probe, and a decrease of the current amplitude. Although the authors reported a lower detection limit, down to 0.05 nM, and a high selectivity to trypsin, this method has some drawbacks. First, the biosensor was based on the peptide substrate labeled by the graphene oxide (GO)–thionine conjugate, which is not available commercially. The peptide–GO–thionine conjugates are more expensive in comparison with the β-casein used in our work. Therefore, this limits the practical application of such an electrochemical sensor. Moreover, unlike the label-free approach presented in our work, the method by Chen et al. cannot monitor the trypsin activity in real time, because this activity was detected only after 2 h of incubation of the trypsin with the peptide-modified electrode. It should be also mentioned that commercially available enzyme-linked immunosorbent assay (ELISA) kits for trypsin possess also high selectivity and sensitivity similar to the work of Chen et al. (down to 0.012 nM) [52]. However, those kits require expensive antibodies, and detection is carried out in several steps. Furthermore, ELISA does not allow monitoring of the kinetics of the trypsin activity. The acoustic sensor developed by us is sufficiently sensitive (LOD of 0.48 ± 0.08 nM) to detect such dangerous diseases as cystic fibrosis, acute pancreatitis, or the acute phase of chronic pancreatitis that are characterized by raised concentration of trypsin in blood in the range of 2.1–71.4 nM [7,8]. In contrast with ELISA, the TSM biosensor is label-free, straightforward, and facile regarding the evaluation of the response. In addition, the TSM method can be used in samples that are not transparent.

### 3.2. Sprectophtometric Assay of Protease Activity

In colorimetric sensor applications, AuNPs are most widely used due to their high stability, facile synthesis, excellent biocompatibility, and strong surface plasmon resonance effect. This effect can be utilized to produce visual color changes in a process termed the colorimetric method [53,54]. Here, we report the results of a simple colorimetric assay based on the optical properties of functionalized AuNPs (Figure 2). The purpose of this study was a comparison of the sensitivity of surface-based (TSM) and volume-sensitive methods of trypsin activity detection. We used a slightly modified version of the method reported by Chuang et al. [26]. However, instead of gelatin, β-casein has been used as a substrate for trypsin digestion. Briefly, for the protease assay, AuNPs were first modified by β-casein and subsequently with MCH. The molecules of MCH are chemisorbed to the AuNPs through a thiol group (-SH) substitution and the hydroxyl group (-OH) exposed on the AuNPs surface enhances the attraction force between AuNPs. Additionally, MCH molecules on the AuNPs act as blockers, while covering the surface area of the AuNPs that are not conjugated with casein and blocking adsorption of the protease on the surface of the AuNPs [26]. The addition of MCH to the AuNPs–β-casein solution led to a color change from wine-red to violet. When trypsin digests the casein at AuNPs/MCH–casein, NPs aggregated due to the removal of the protective layer of casein and the color change from violet to blue occurred within minutes; then, the solution became colorless.

The absorption spectra of AuNPs in the absence of β-casein (black curve), presence of β-casein (red curve), presence of β-casein and MCH (blue curve), and AuNPs with chemisorbed MCH (magenta curve) are shown in Figure 6. The absorption peak of pure AuNPs is centered at 520 nm as expected. This indicates that the gold colloids are not aggregated but well dispersed as individual particles [55]. After the modification of AuNPs with β-casein, the position of the maximum absorption of AuNPs shifted from 520 to 525 nm, which indicates the formation of bioconjugates [55]. The shift is identical with those reported in [26] for AuNPs modified by gelatin. The red shift in the position of the plasmon absorption band is produced by a perturbation in the dielectric constant around the nanoparticles due to the chemisorption of β-casein molecules [56]. No significant broadening of the spectrum was observed after the β-casein adsorption process, which indicates that the separation distance between AuNPs is higher than their radii, and that AuNPs do not experience aggregation into larger nanoparticles upon the adsorption of β-casein [55]. Further modification with MCH resulted in a significant red shift around 60 nm accompanied by the broadening of the spectrum. This broadening is indicative of an aggregation of nanoparticles This is due to the replacement of the β-casein protective layer with MCH, which in turn makes the nanoparticles closer to each other [57,58]. The modification of AuNPs with MCH resulted in a significant red shift, indicating strong aggregation of the nanoparticles.

Additionally, a less expressed maximum at 280 nm is observed after the modification of AuNPs by β-casein. This is due to the absorption of β-casein’s amino acids at this wavelength. The amplitude of this peak decreases after the chemisorption of MCH, which is probably due to the removal of weakly adsorbed casein molecules from the surface of AuNPs. Furthermore, we carried out a quantitative analysis of trypsin activity via the UV-vis spectroscopy method. For this purpose, trypsin was added to the AuNPs solution. We recorded the changes of absorbance spectra of the AuNPs suspension during the trypsin cleavage at 0 min, 0.01 min, 15 min, 30 min, 45 min, and 60 min. Figure 7 illustrates the changes in spectra over time in a 10 nM concentration of trypsin. A substantial red shift (up to 640 nm) of the spectra and a decrease in absorbance with time was observed at this concentration of trypsin, due to trypsin-induced aggregation caused by the cleavage of the AuNPs’ protective shell as well as the MCH induced increase of attractive force between the AuNPs. Moreover, the absorbance spectra showed a decrease in the absorption spectra at 280 nm when the AuNPs/MCH–casein was digested by trypsin. It can also be seen that the absorbance decreased with time. The absorbance also started to decrease after maximum shifting. Our results are in good agreement with those previously reported by Chuang et al., whose work served as our inspiration to design a colorimetric assay based on an AuNPs/MCH-protein platform. Chuang et al. demonstrated that protein modified AuNPs aggregation after treatment with protease can be successfully monitored via the red shift of absorption spectra [26].

Trypsin at concentrations ranging from 0.1 to 10 nM was used in the study to estimate the detection limit of the optical AuNPs assay. In order to construct the calibration curve, we have plotted the changes in relative values of absorbance measured at around 640 nm after 15 min of trypsin exposure against the trypsin concentration (Figure 8a). As in the case of the analysis of trypsin activity via the TSM method, we were able to use an inverse Michaelis–Menten (MM) model expressed by Equation (3) to analyze the obtained calibration curve for trypsin activity in volume.

The fitting of calibration plots with the MM model yielded *v_max_* = 14.98 ± 0.81% and K_M_ = 0.56 ± 0.10 nM. As was already mentioned, in the inverse MM model, the roles of the enzyme and substrate are swapped, and the concentration of the enzyme is changed while the substrate is presented in excess.

To calculate the LOD, we used only part of the calibration curve from 0 to 5 nM, where the dependence was almost linear. The obtained LOD was 0.42 ± 0.03 nM, according to the rule 3.3 (SD)/S, where SD is the standard deviation of the sample with the lowest concentration and S is slope calculated from the fit of the linear part of the calibration curve [59]. The results are shown in Figure 8b.

It is interesting to compare the properties of the AuNPs assay and the TSM method used to detect trypsin activity (Table 1). On one hand, both methods successfully detected protease activity at the sub-nM level, within a similar time range in a real-time mode. However, it should be noted that a major drawback of the AuNPs assay is that the method is of limited application in a turbid medium. On the other hand, unlike the TSM method, detection using the AuNPs assay can be carried out in only one step, as the signal detection simply involves the direct measurement of the absorbance values at A_640_. It is also interesting to compare the reverse Michaelis–Menten constants for both methods. As can be seen in Table 1, a lower K_M_ value has been obtained for the AuNPs-based colorimetric assay. This can be attributed to trypsin’s better access to the β-casein substrate. Certainly, the β-casein layer is formed at MUA monolayers by covalent binding of the casein hydrophilic amino groups. Thus, the cleavage sites are closer to the quartz crystal surface with limited access to the trypsin. In addition, due to covalent binding of casein molecules at the self-assembled MUA, the casein layer is compactly packed, which creates additional restriction of access of trypsin to the cleavage sites. A similar conclusion was also obtained for chymotrypsin detection [57]. In contrast, at AuNPs, the casein is physically adsorbed at the gold surface. This means that casein molecules are randomly oriented, which make the access of trypsin to the casein cleavage sites more advantageous.

We should also mention that in contrast with the colorimetric method, the acoustic TSM technique is sensitive to air bubbles presented in the sample and to the pressure changes caused by handling of the flow cell. Air bubbles are more prone to growth at the hydrophilic interface, which likely altered to a hydrophobic case upon the adsorption of the casein layer. Therefore, special care can be taken in avoiding this effect, for example by degassing the sample before starting the experiments.

Finally, we briefly discuss the most often used techniques employed for trypsin detection. The advantages and disadvantages of these techniques as well as their LOD are summarized in Table 2. Nowadays, researchers’ efforts are focused on the development of simple and rapid biosensors for the sensitive determination of trypsin because traditional methods such as enzyme-linked immunosorbent assay (ELISA), gelatin-based film technique, and high-performance liquid-chromatography (HPLC) are time-consuming and require specialized instruments and trained personnel [60]. Moreover, those methods do not allow for the monitoring of the kinetics of protease activity. Recently, many efforts have been reported regarding trypsin determination. Several biosensors based on fluorescent, electrochemical, and colorimetric methods have been developed to detect trypsin [15,26,60,61]. Fluorescence-based homogeneous assays are the most popular ones for trypsin activity monitoring due to their simple processes, high sensitivity, and convenient operation. These methods usually need peptide-based molecular probes containing fluorochrome and quencher pairs to monitor specific proteases by fluorescence resonance energy transfer. Nevertheless, these labeled fluorogenic substrates are expensive and are difficult to synthesize [62]. Colorimetry is another method reported for the detection and screening of trypsin [63]. It is the simplest, less expensive, and most widely used method. It can be directly observed with the naked eye or accurately quantified via UV-vis spectrophotometer. Unfortunately, the colorimetric method is limited to only optically transparent liquids. Electrochemical methods are rather sensitive. However, they require the conjugation of a specific peptide substrate by redox probes, longer incubation time with protease, and cannot be used for measuring the kinetics of enzyme reaction [51]. Acoustic methods are among the most effective and promising approaches for the detection of trypsin activity. Their advantage lies in high sensitivity, which reaches levels comparable to state-of-the-art techniques such as ELISA. Since most of the biochemical samples are acoustically transparent, measurements can be performed in a wider range of solutions without the need for a chemical probe, as well as in opaque and high-concentration samples that are difficult to measure with optical methods. Most recently, we successfully demonstrated the feasibility of a volume-sensitive acoustic method for the detection of proteolytic activity of trypsin [64]. However, it is worth noting a possible limitation of the proposed acoustic methods: namely, the influence of air bubbles and temperature stability. This limitation must be addressed in future research.

## 4. Conclusions

We have shown that β-casein forms a stable monolayer via an 11-mercaptoundecanoic acid (MUA) cross-linker at the gold surface of a piezoelectric transducer. The TSM sensor based on a β-casein layer enabled a detection limit of 0.48 ± 0.08 nM for trypsin. The cleavage of β-casein resulted in an increase of resonant frequency and a decrease of motional resistance. Furthermore, we compared the results obtained by the TSM method with a colorimetric assay for quantifying trypsin activity in a volume. This assay was based on AuNPs modified by β-casein and MCH and on the phenomena of surface plasmon resonance (SPR) and yielded a detection limit of 0.42 ± 0.03 nM, which is comparable with the LOD obtained from TSM experiments. We also analyzed the Michaelis–Menten constants, K_M_, for reverse enzymatic reaction and showed that the K_M_ value for the colorimetric assay (0.56 ± 0.10 nM) is lower in comparison with that obtained in the case of the TSM method (0.92 ± 0.44 nM). This has been explained by better access of trypsin to the β-casein in a volume. The TSM method is useful for the study of the kinetics of the protease’s activity, which is not possible via conventional ELISA or HPLC methods. The obtained results can be considered as a first step toward the application of a TSM sensor and colorimetric assays based on β-casein for the label-free detection of trypsin activity. For practical application in medical diagnostics, both acoustic and optical methods need additional validation in complex biological fluids such as blood or blood plasma. In addition, the sensitivity of the TSM method can be improved by the application of hydrophobic substrates for casein immobilization. We anticipate in this case that the detection limit can be improved at least five times. The improved sensitivity of detection is important for working with diluted bilogical samples in order to minimize the matrix effect.

## Figures and Tables

**Figure 1 biosensors-11-00117-f001:**
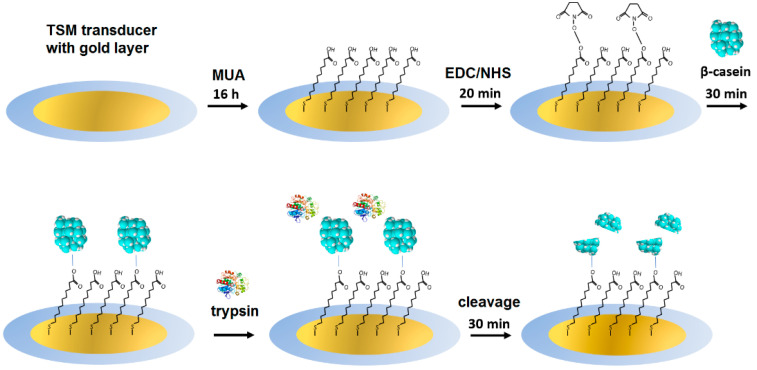
The scheme for modification of the gold layer on a TSM transducer and the cleavage of β-casein by trypsin.

**Figure 2 biosensors-11-00117-f002:**
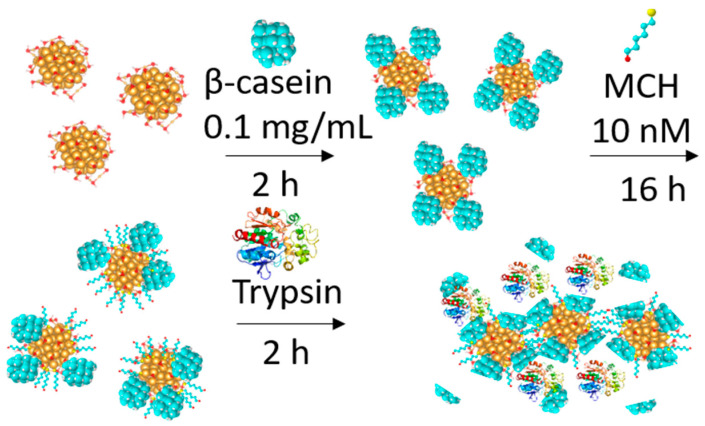
The scheme for modification of gold nanoparticles (AuNPs) by β-casein and by 6-mercapto-1-hexanol (MCH) as well as the cleavage of β-casein by trypsin. Before enzymatic digestion, functionalized AuNPs were stable due to steric stabilization. After the AuNPs were subjected to protease cleavage, the casein was removed from the surface of AuNPs/MCH/β-casein. This caused the destabilization of the NPs, followed by their aggregation.

**Figure 3 biosensors-11-00117-f003:**
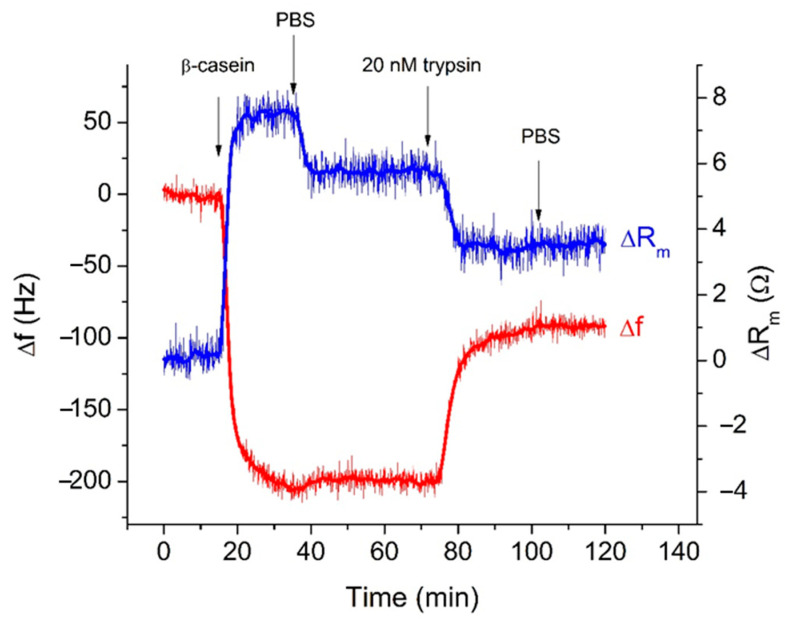
Typical kinetics of the changes of resonant frequency, Δf, and motional resistance, ΔR_m_, of the thickness-shear mode (TSM) transducer for various modifications. The additions of β-casein, trypsin, and washing of the surface by phosphate-buffered saline (PBS) are shown by arrows.

**Figure 4 biosensors-11-00117-f004:**
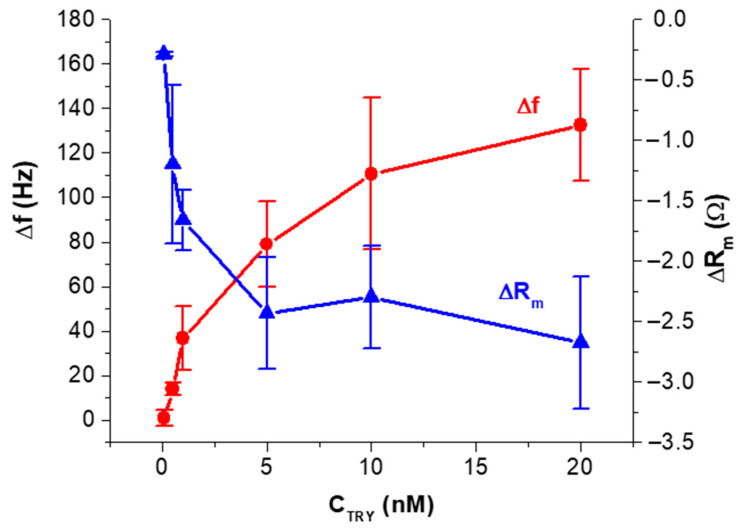
Plots of changes of frequency, Δf, and motional resistance, ΔR_m_, vs. trypsin concentration (C_TRY_). Statistically, a value for the standard deviation was obtained from three independent experiments at each trypsin concentration.

**Figure 5 biosensors-11-00117-f005:**
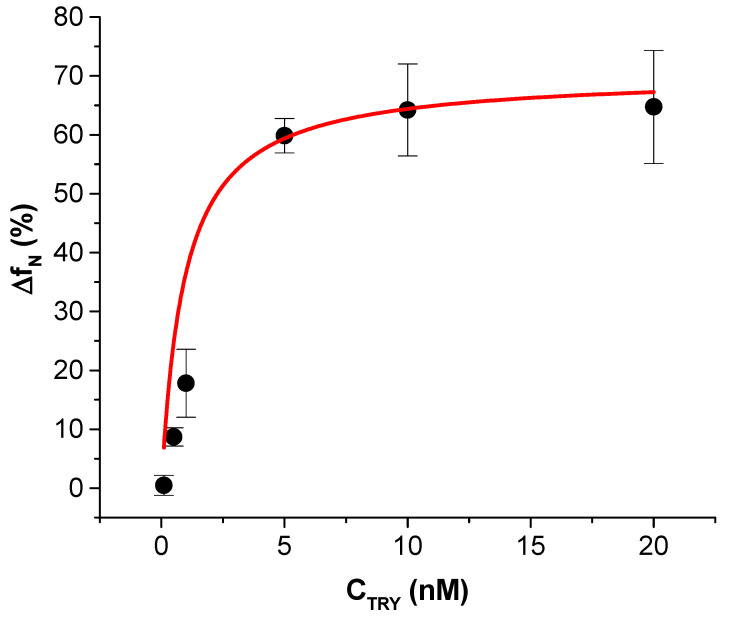
Plot of the normalized changes of the resonant frequency Δf_N_ vs. trypsin concentrations, C_TRY_. Standard deviation values are obtained from three independent experiments. The red line is the fit according to the Langmuir isotherm (Equation (2)) with accuracy R^2^ = 0.99.

**Figure 6 biosensors-11-00117-f006:**
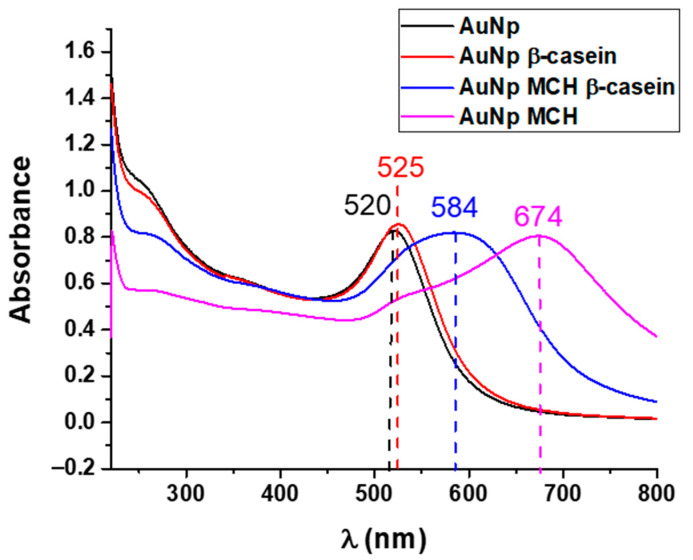
UV-vis absorption spectra of gold nanoparticles (AuNPs): bare (black), modified by β-casein (red), and subsequently modified by 6-mercapto-1-hexanol (MCH) (blue) as well as AuNPs modified by MCH (magenta).

**Figure 7 biosensors-11-00117-f007:**
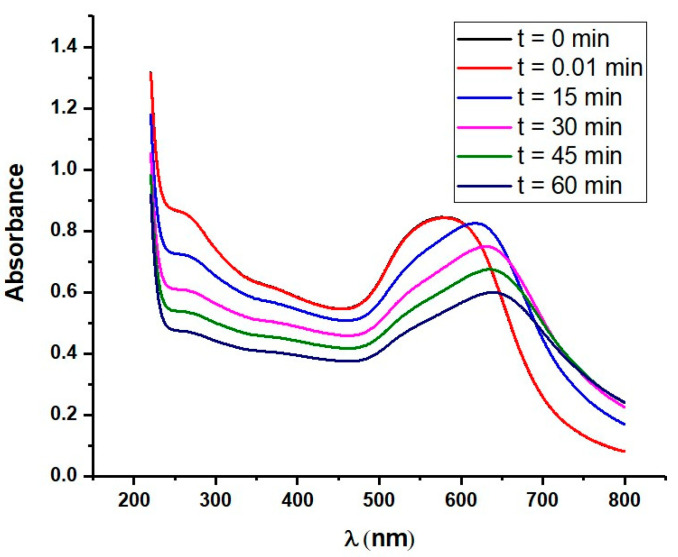
UV-vis absorption spectra of β-casein and MCH-conjugated AuNPs treated with 10 nM trypsin at different time points. Note that at time 0 and 0.01 min, the spectra are almost identical.

**Figure 8 biosensors-11-00117-f008:**
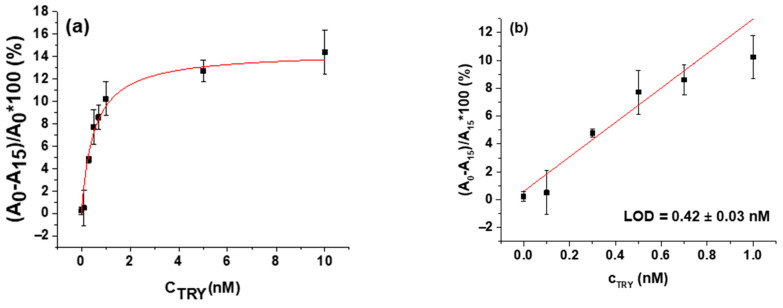
Calibration plots of colorimetric assay. (**a**) Changes in relative values of absorbance after β-casein and MCH functionalized gold nanoparticles (AuNPs/MCH-β-casein) exposure to trypsin (A_0_—exposure time 0 min, A_15_—exposure time 15 min.) vs. concentration of trypsin (C_TRY_). Symbols are experimental data, and the red line is the best fit of Equation (3). (**b**) Linear part of the calibration curve for calculation of the limit of detection (LOD). Values are means ± SD (*n* = 3). Red line is the linear regression fit.

**Table 1 biosensors-11-00117-t001:** Comparison of TSM biosensor and AuNPs platform (colorimetric biosensor) used for detection of trypsin.

Parameters	TSM Biosensor	AuNPs Assay
Detection time	30 min	30 min
K_M_	0.92 ± 0.44 nM	0.56 ± 0.10 nM
Detection limit	0.48 ± 0.08 nM	0.42 ± 0.03 nM
Signal detection	Acoustic wave at surface	UV-vis absorbance in a volume

**Table 2 biosensors-11-00117-t002:** Comparison of the most used analytical methods for trypsin determination.

Method	Advantages	Disadvantages	LOD, nM	References
ELISA	High selectivity and sensitivity	Requires expensive antibodies, the kinetics of trypsin activity cannot be measured	0.012	[42]
Fluorescent assay	High sensitivity, operates in real-time mode	Fluorogenic substrates are expensive and difficult to be synthesized.	3.8–29	[15,61]
Colorimetric assay	Simple, inexpensive, and sensitive, enables real-time detection of trypsin activity	Limited to only optically transparent liquids	0.19 0.42 ± 0.03	[63] This work
Electrochemical sensor	High sensitivityy	Necessity to use peptide substrate conjugated with graphene oxide and thionine	0.05	[51]
Acoustic TSM sensor	High sensitivity, capable of real-time monitoring of kinetics of the trypsin mediated cleavage	Measurements are sensitive to air bubbles presented in the sample	0.2 0.48 ± 0.08	[23] This work
High-resolution ultrasonic spectroscopy	High sensitivity, capable of real-time monitoring of kinetics of the trypsin mediated cleavage	Measurements are sensitive to air bubbles presented in the sample	~1.0	[64]

## Data Availability

Not applicable.

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
