# Peer review of "Detection of Sub-Nanomolar Concentration of Trypsin by Thickness-Shear Mode Acoustic Biosensor and Spectrophotometry"

_biosensors, 2021, doi:10.3390/bios11040117_

Round 1

Reviewer 1 Report

In this manuscript, the authors report results of detection of nanomolar concentration of trypsin using acoustic biosensor and spectrophotometry. Manuscript can not be accepted in the present form and need extensive experimental work and thus I recommend its rejection for the following reasons:

  • The use of gold electrode modified with a substrate of trypsin was already reported by Chen, G., Shi, H., Ban, F., Zhang, Y., & Sun, L. (2015). Determination of trypsin activity using a gold electrode modified with a nanocover composed of graphene oxide and thionine. Microchimica Acta, 182(15-16), 2469–2476.doi:10.1007/s00604-015-1601-x. The authors of this published paper succeed to reach a concentration of 0.05 nM of trypsin much lower than the value reported in this manuscript.
  • As indicated in Figures 3 and 5, no linearity was observed. Indeed, concentrations of 5, 10, and 20 nM give a similar response.
  • The authors compared their obtained results with a colorimetric method based on gold nanoparticles. This colorimetric method is not a standard or reference method and thus needs to be validated. However, there are many commercially available kits (Sigma-Adrich, Abcam,…) based on colorimetric methods which are widely employed in different labs in the world.
  • The large difference in values of Km (24.38 and 0.51 nM) can not be explained only by “better access to the casein substrate” as claimed by the authors (lines 436 and 437).

Author Response

We are grateful to the reviewer for helpful comments that allowed us to improve the manuscript. We performed additional experiments and analysis of the results. The reviewer's comments are addressed either in the revised paper or in the responses below. All changes made in the revised manuscript are highlighted by yellow.

Comment 1. The use of gold electrode modified with a substrate of trypsin was already reported by Chen, G., Shi, H., Ban, F., Zhang, Y., & Sun, L. (2015). Determination of trypsin activity using a gold electrode modified with a nanocover composed of graphene oxide and thionine. Microchimica Acta, 182(15-16), 2469–2476.doi:10.1007/s00604-015-1601-x. The authors of this published paper succeed to reach a concentration of 0.05 nM of trypsin much lower than the value reported in this manuscript.

Response: The article by Chen et al. (2015) reports detection of trypsin activity based on an electrochemical method. Although the authors reported a lower detection limit, this method has some drawbacks making the assay more complicated in comparison with those reported in our work. First, the biosensor developed by Chen et al. was based on the peptide substrate labeled by graphene oxide (GO)-thionine conjugate. The cleavage of the peptide substrate by trypsin has been monitored by decreased redox current due to removal of the thionine redox label. However, unlike to the label-free methods presented in our work those by Chen et al. can not monitor the trypsin activity in a real time, because it was detected only after 2 hours of incubation of the peptide-modified working electrode with the trypsin. Second, the peptide-GO-thionine conjugates are not available at thee market and the preparation of the sensor is more laborious and time consuming in comparison with those based on β-casein layers used in our work. This limits the practical application of such an electrochemical sensor. It should be also mentioned that commercially available enzyme-linked immunosorbent assay (ELISA) also allowing high selectivity and sensitivity for trypsin detection down to 0.012 nM. However, the ELISA kit requires expensive antibodies, and is carried out in several steps. Moreover, ELISA does not allow monitoring of the kinetics of the trypsin activity.

          The acoustic and colorimetric sensors described in our manuscript are sufficiently sensitive to detect serious diseases such as cystic fibrosis, acute pancreatitis or the acute phase of chronic pancreatitis. These types of pancreatic diseases are associated with the increased trypsin level in serum of patients. Typically, the range of 2.1- 71.42 nM trypsin is observed during these diseases [Sharma, H.; Vyas, R.K.; Vyas, S. Role of Serum Trypsin Level in Diagnosis and Prognosis of Pancreatitis and Compared with Healthy Subjects of Rajasthan. Amer. J. Biochem. 2018, 8, 93-99. doi: 10.5923/j.ajb.20180805.02; Heinrich, H.C.; Gabbe, E.E.; Ičagić, F. Immunoreactive serum trypsin in diseases of the pancreas. Klin Wochenschr. 1979, 57, 1237-1238. doi: 10.1007/BF01489252]. Therefore, both methods reported in our paper are sufficiently sensitive and useful for medical diagnosis. Moreover, the acoustic method is suitable for the study of the kinetics of the protease’s activity, which is not possible by above mentioned electrochemical sensor and ELISA assay. In addition, both acoustic and colorimetric methods are label free and allowed straightforward and rapid detection of the trypsin. The explanation presented above has been included in the revised manuscript.

Comment 2: As indicated in Figures 3 and 5, no linearity was observed. Indeed, concentrations of 5, 10, and 20 nM give a similar response.

Response: The cleavage of the β-casein by trypsin is an enzymatic reaction that follows Michaelis-Menten behavior. The plot of the reaction rate as a function of the substrate or enzyme concentration (in the case of excess substrate concentration) can not be linear, but it is well fitted by Langmuir isotherm, that can be used as a calibration curve. The newly performed experiments based on using more effective cleaning method of the gold layers demonstrated different values of the resonant frequency and motional resistance in the range of 0.1-5 nM of trypsin. At higher trypsin concentration the saturation took place which agrees well with colorimetric assay (see Figures 4, 5  and 8a of revised manuscript).

Comment: The authors compared their obtained results with a colorimetric method based on gold nanoparticles. This colorimetric method is not a standard or reference method and thus needs to be validated. However, there are many commercially available kits (Sigma-Aldrich, Abcam,…) based on colorimetric methods which are widely employed in different labs in the world.

Response: The colorimetric method that we used has been inspired by work of Chuang et al. (2010) discussed in the manuscript. The use of gold nanoparticles improves the colorimetric response. The commercially available kits indicated by reviewer are based on the formation of pNA from a chromogenic substrate. These assays, however, requires around 1-2 hours with sensitivity in the μM range of trypsin, which is less in comparison with colorimetric method based on AuNPs reported in our paper. We, however, agree with reviewer that further validation of colorimetric method is required, especially in the detection of trypsin in the real samples. We therefore included suggestions for future applications of the colorimetric and acoustic methods in Conclusion section of the revised manuscript.

Comment 4: The large difference in values of KM (24.38 and 0.51 nM) can not be explained only by “better access to the casein substrate” as claimed by the authors (lines 436 and 437).

Response: We agree with this comment and performed additional TSM experiments in which used fresh trypsin like in the case of colorimetric assay as well as more effective method of the cleaning of the gold surface of TSM transducers. In addition, we analyzed the sensor response in respect of certain variations in the frequency changes caused by formation of β-casein layers. The calibration curve that represent plot of the normalized frequency changes ΔfN vs. concentration of trypsin (CTRY) (ΔfN=(ΔfTRY/Δfcasein)x100(%), where ΔfTRY are changes in frequency following addition of trypsin at certain concentration of the protease and Δfcasein are changes in frequency caused by formation of β-casein layer) is presented on Figure 5 of the revised manuscript. The newly obtained KM value (0.92±0.44 nM) now does not differ dramatically from those obtained in colorimetric assay (0.56±0.10 nM) and can be attributed to the different access of the trypsin to the casein cleavage site.

Reviewer 2 Report

In this manuscript, the authors propose two methods to detect trypsin detection. Here are some main criticisms of the paper. I would not recommend it for publication in Biosensor in its current form.

  1. Crystal to crystal variation in MUA coating, casein coating? Are there ways to normalize against this variation?
  2. The fig-2 caption should be delf? Please be careful about using -ve numbers for frequency. This is a recurring minor issue throughout the manuscript.
  3. Figure3: how the frequency is related to the fraction of total protease activity? Why the Hill equation is used for fitting? What is the Hill coefficient? Are the hydrolysis events not independent of each other? Is there cooperativity involved?
  4. Figure-4 is poorly explained and fails to connect to any physical variable related to the enzymatic hydrolysis
  5. Please explain more clearly how df/dt is calculated? At which time point? This is not saturating. Why the authors do not go to a higher concentration of Trypsin? Typically, in enzyme kinetics, MM-equation Vmax represents the saturation level of activity. And the Km is defined by the concertation of the enzyme needed to reach a characteristic activity level. Here the physical meanings are unclear. The authors define Vmax as: “represents the maximum rate of enzyme reaction” which is different from a classical definition. Thus, the Km is also different. The authors should make this point very clear, to avoid confusion.
  6. The authors should show a characteristic absorption spectrum of aggregated GNP. Presumably, it should be similar to AU-MCH spectra? How do the spectra look like at AU-MCH only, AU-MCH, and freely floating b-casein?
  7. Why is there a blue-shift in 0,0.01 min spectra compared to the rest of the absorption spectra?
  8. 7 similar issues: A is calculated at which wavelength? This, however, follows a more traditional definition of enzyme-kinetics. The difference is absorbance represents the fraction of molecules hydrolyzed. Thus, the KM has a different meaning!
  9. Table2: As stated above, the comparisons are not the correct parameters.
  10. The authors should discuss in detail why, given the relatively high LOD, and accessibility of trypsin issue for the acoustic wave experiments, why this is a useful technique?

Author Response

We are grateful to the reviewer for helpful comments that allowed us to improve the manuscript. We performed additional experiments and analysis of the results. The reviewer's comments are addressed either in the revised paper or in the responses below. All changes made in the revised manuscript are highlighted by yellow.

Comment 1: Crystal to crystal variation in MUA coating, casein coating? Are there ways to normalize against this variation?

Response: A calibration plot of the degree of the cleavage of β-casein layers as a function of trypsin concentration can be constructed to normalize certain variations in Δf values that depended on the properties and cleanliness of the quartz crystals, MUA and β-casein coating and stability. The degree of casein cleavage represented by the changes of the normalized frequency, ΔfN (in %) by trypsin can be calculated as follows: ΔfN=(ΔfTRY/Δfcasein)x100% where ΔfTRY is the frequency changes corresponded to the cleavage of casein layer after incubation with certain concentration of trypsin and Δfcasein is the frequency change corresponded to the adsorption of β-casein at MUA layer before trypsin addition. The plot of ΔfN vs. CTRY has been used as a calibration curve in revised manuscript (Figure 5).

Comment 2: The fig-2 caption should be delf? Please be careful about using negative numbers for frequency. This is a recurring minor issue throughout the manuscript.

Response: We agree with this comment and change f to Δf on the coordinates of Figures 3 to 5 at revised manuscript. The negative numbers for frequency changes are indication of the decrease in initial frequency of quartz crystal due to the formation of β-casein layer. Note, that in the revised manuscript we report new experiments and evaluated the cleavage of β-casein by trypsin using frequency and motional resistance changes in a steady-state conditions. This allowed more correct comparison of TSM and colorimetric data.

Comment 3: Figure 3: how the frequency is related to the fraction of total protease activity? Why the Hill equation is used for fitting? What is the Hill coefficient? Are the hydrolysis events not independent of each other? Is there cooperativity involved?

Response: The presence of trypsin in solution results in an increase of resonant frequency. The increase of frequency corresponds to a decrease of thickness or mass of the β-casein layer (please see Equation 1 in Materials and Methods) due to the cleavage of β-casein by trypsin. This gives us possibility to estimate overall activity of trypsin. The Hill equation has been previously used in a context other than the original concept known in the field of biochemistry. Due to noise presented in the measurement data calculation of reaction rate by first order differentiation of curves could not lead to useful outcome. Therefore, the Hill equation was applied to fit each dataset corresponding to a certain trypsin concentration. Furthermore, the fitted curves were numerically differentiated at initial time of enzyme reaction, where the frequency changes were close to the linear. The hydrolysis reaction of β-casein obeys Michaelis – Menten kinetics and there are no cooperative hydrolysis reactions. Hill coefficient was 1.02 after fitting the curves on Figure 3. However, for practical application of trypsin detection it is more convenient to analyze the changes in resonant frequency in steady-state conditions. Also, the normalized frequency changes that consider variation in the frequency changes following adsorption of β-casein should be taken into account. Therefore, we performed this analysis based on newly performed experiments and presented this in revised manuscript.

Comment 4:  Figure-4 is poorly explained and fails to connect to any physical variable related to the enzymatic hydrolysis

Response: In the revised manuscript we report newly performed experiments and analysis of the cleavage of β-casein layer by trypsin using changes in resonant frequency and motional resistance. This allowed more correct comparison with colorimetric data. Based on the frequency and motional resistance changes we also analyzed possible viscosity contribution of the β-casein cleavage into the frequency changes and showed that those changes are mostly due to mass contribution.

Comment 5: Please explain more clearly how df/dt is calculated? At which time point? This is not saturating. Why the authors do not go to a higher concentration of Trypsin? Typically, in enzyme kinetics, MM-equation Vmax represents the saturation level of activity. And the Km is defined by the concentration of the enzyme needed to reach a characteristic activity level. Here the physical meanings are unclear. The authors define Vmax as: “represents the maximum rate of enzyme reaction” which is different from a classical definition. Thus, the Km is also different. The authors should make this point very clear, to avoid confusion.

Response: df/dt values were obtained from the numerically differentiated Hill curves presented on the Figure 3 differentiated at the time point corresponding to at initial time of enzyme reaction, where the frequency changes were close to the linear. The calculated df/dt represents a maximum value of the differentiated Hill curve. This range of concentration for trypsin in our study was applied because at this concentration trypsin is usually detected in serum samples of patient with pancreatic diseases. Here, KM represents Michaelis-Menten constant from inverse MM equation where enzyme and substrate are reversed. Vmax is maximum rate achieved by the system, happening at saturating enzyme concentration. Therefore, KM indicates at what concentration the enzyme reaches half-rate. However, unlike the classical MM equation this represents saturation of enzyme.

          As we mentioned above in the response to the Comment 3, we used normalized frequency changes for construction of calibration curve and for determination of the KM value. The newly obtained results and analysis allowed more correct comparison of the TSM and colorimetric data.

Comment 6: The authors should show a characteristic absorption spectrum of aggregated GNP. Presumably, it should be similar to AU-MCH spectra? How do the spectra look like at AU-MCH only, AU-MCH, and freely floating b-casein?

Response: There are different spectra of aggregated GNP depending on the cause of the aggregation and GNP distribution in the sample. We added Au-MCH spectra for comparison in Figure 6 of revised manuscript. Addition of free floating β-casein does not change the spectrum.

Comment 7: Why is there a blue-shift in 0,0.01 min spectra compared to the rest of the absorption spectra?

Response: There is no significant blue shift. The spectra are almost identical, and any difference is related to a difference in measurement. We added note on this in the caption of Figure 7 of the revised manuscript.

Comment 8: similar issues: A is calculated at which wavelength? This, however, follows a more traditional definition of enzyme-kinetics. The difference is absorbance represents the fraction of molecules hydrolyzed. Thus, the KM has a different meaning!

Response: Absorbance, A, is calculated at absorbance maximum. In both cases KM represents Michaelis-Menten constant from inverse M-M equation where enzyme and substrate are reversed. In both cases it suggests at what concentration the enzyme reaches half-rate. However while in normal M-M equation this represents saturation of enzyme, in inverse M-M equation it is saturation of the substrate.

Comment 9: Table 2: As stated above, the comparisons are not the correct parameters.

Response: The detection time, detection limit and KM values are convenient parameters for comparison of the methods of detection of trypsin by various methods. We therefore mean that this table (Table 1 in revised manuscript) can be useful for the purpose of the comparison.

Comment 10. The authors should discuss in detail why, given the relatively high LOD, and accessibility of trypsin issue for the acoustic wave experiments, why this is a useful technique?

Response: The advantages of TSM biosensor was discussed in the revised manuscript.

Reviewer 3 Report

Melikishvili et al. reported a method of measuring protease activity using a colorimetric and acoustic wave-based biosensor approach. These techniques can be classified as fast techniques with a detection time of 30 minutes compared to other techniques, and have the advantage of being able to measure the detection limit up to sub-nanomolar. Although the background of the study and the approach of the study were well organized, I judged that some of the contents were not arranged appropriately for each section. Meanwhile, in order for manuscript to be accepted, the following concerns should be addressed.

Introduction
1.    Line 71-103: “In this work, we designed an analytical method based on the TSM biosensor for the ………. with the naked eye or monitored through the absorbance change at 640 nm.” I think these sentences are a mix of what will be covered in the discussion. I suggest that the author write the background and purpose of this study more concisely in Introduction.

Materials and method 
1.    I suggest excluding the molecular weight of the materials. Less important information hinders readability.
2.    The author purchased Trypsin from Sigma-aldrich and used it in the experiment without further purification. If some of the Trypsin are inactive, some of the experimental results may show different values. Accordingly, the authors need a clear explanation of the state and activity of commercial Trypsin used in this experiment.
3.    Line 165: “For proteolysis measurements, solutions with various concentrations of trypsin in PBS were flowed ...” Author should provide the exact concentration of trypsin.
4.    Line 189-190: “Absorbance was measured by UV-1700 spectrophotometer(Shimadzu, Kyoto, Japan).” It should be added whether the measurements were taken in the wavelength range and the experimental temperature. 

Result and Discussion
1.    For the equations used to analyze the experimental results, it is appropriate to move to the Method section rather than Results and discussion. Also, in the case of the fitting parameter in Table 1, it is judged that it is more suitable for the method than the result.
2.    Figure3: Why are the different concentrations of trypsin measured over different time ranges?
3.    Figure 4. The author needs to increase the line symbol indicating concentration. 
4.    Figure 5. In the plot for δf/δt as a function of Trypsin concentration, the value of 20 nM Trypsin is not consistent across the entire reaction. The author should clarify this part. 5. Figure 7. T=0min and t=0.01 min are almost identical. As this is difficult to observe clearly in the figure, I suggest adding this to the figure caption. 6. Figure 8: Tables are not arranged in Figures 8a and 8b, and the text size in the figure is different. I suggest changing it to the same size.
5.    I suggest author to describe future research application that can apply the current technique to improve the quality of manuscript.

Author Response

We are grateful to the reviewer for helpful comments that allowed us to improve the manuscript. We performed additional experiments and analysis of the results. The reviewer's comments are addressed either in the revised paper or in the responses below. All changes made in the revised manuscript are highlighted by yellow.

Comment 1: Line 71-103: “In this work, we designed an analytical method based on the TSM biosensor for the ………. with the naked eye or monitored through the absorbance change at 640 nm.” I think these sentences are a mix of what will be covered in the discussion. I suggest that the author write the background and purpose of this study more concisely in Introduction.

Response: Thank you for this comment. The Introduction was updated and rearranged as requested.

Materials and method 
Comment 2: I suggest excluding the molecular weight of the materials. Less important information hinders readability.

Response: Molecular weights of materials were removed as suggested by reviewer.

Comment 3: The author purchased Trypsin from Sigma-Aldrich and used it in the experiment without further purification. If some of the Trypsin is inactive, some of the experimental results may show different values. Accordingly, the authors need a clear explanation of the state and activity of commercial Trypsin used in this experiment.

Response: According to the certificate of analysis provided by Sigma-Aldrich, trypsin used in experiments was of high purity (≥90% protein) with enzymatic activity ≥ 7500 BAEE units/mg solid. Therefore, no further purification was needed.

Comment 4: Line 165: “For proteolysis measurements, solutions with various concentrations of trypsin in PBS were flowed ...” Author should provide the exact concentration of trypsin.

Response: The concentration of trypsin used was 0.1, 0.5, 1, 5,10 and 20 nM. The text has been changed accordingly.

Comment 5: Line 189-190: “Absorbance was measured by UV-1700 spectrophotometer (Shimadzu, Kyoto, Japan).” It should be added whether the measurements were taken in the wavelength range and the experimental temperature. 

Response: We added wavelength range (220-800 nm) and temperature at which the experiments were performed (20 0C) at page 4 of revised manuscript

Result and Discussion

Comment 6: For the equations used to analyze the experimental results, it is appropriate to move to the Method section rather than Results and discussion. Also, in the case of the fitting parameter in Table 1, it is judged that it is more suitable for the method than the result.

Response: The equations were moved to Method section. In the revised manuscript we performed additional experiments and analysis. We used the changes of resonant frequency and motional resistance for evaluation of the sensor response and for estimation of the reverse Michaelis-Menten constant KM. We also normalized the changes in frequency on a certain variation in the frequency changes following adsorption of the casein onto MUA layers. Better correlation between TSM and colorimetric data has been obtained using this new approach.

Comment 7:  Figure 3: Why are the different concentrations of trypsin measured over different time ranges?

Response: The Results and Discussion section concerning TSM data has been considerably updated based on new experiments and data analysis.

Comment 8:  Figure 4. The author needs to increase the line symbol indicating concentration. 

Response: The symbols were increased in all newly prepared figures.

Comment 9: Figure 5. In the plot for δf/δt as a function of Trypsin concentration, the value of 20 nM Trypsin is not consistent across the entire reaction. The author should clarify this part.

Response: This plot has been replaced by a plot of the changes of the normalized frequency ΔfN vs. trypsin concentration, which more correctly reflects the cleavage of casein by trypsin. Newly prepared calibration curve (Figure 5) has been used also for evaluation of KM constant.

Comment 10. Figure 7. t=0min and t=0.01 min are almost identical. As this is difficult to observe clearly in the figure, I suggest adding this to the figure caption.

Response: We added a note to the Figure 7 caption.

Comment 11. Figure 8: Tables are not arranged in Figures 8a and 8b, and the text size in the figure is different. I suggest changing it to the same size.

Response: We changed the text size in the Figure 8 as suggested by reviewer.

Comment 12: I suggest author to describe future research application that can apply the current technique to improve the quality of manuscript.

Response: We added possible future applications in the Conclusion section.

Round 2

Reviewer 1 Report

The manuscript was much improved. The authors answer all my remarks and comments.

Reviewer 2 Report

The authors have implemented most of my concerns and have improved the quality of the paper significantly. I recommend it for publication in biosensor.

Reviewer 3 Report

The author has address all my concerns, and I only ask for corrections to one minor points below. I recommend to accept the revised manuscript for publication. 

Minor
In author contrbution, the name expression of authors must be unified.